# Feature ranking and network analysis of global financial indices

**Mahmudul Islam Rakib**, **Md. Javed Hossain, Ashadun Nobi** *

Department of Computer Science and Telecommunication Engineering, Noakhali Science and Technology University, Sonapur, Noakhali, Bangladesh

* ashadunnobi_305@yahoo.com

**Data Availability Statement:** All data are fully available without restriction. The data underlying the results presented in the study are available from Yahoo Finance (https://finance.yahoo.com/world-indices).

## Abstract

The feature ranking method of machine learning is applied to investigate the feature ranking and network properties of 21 world stock indices. The feature ranking is the probability of influence of each index on the target. The feature ranking matrix is determined by using the returns of indices on a certain day to predict the price returns of the next day using Random Forest and Gradient Boosting. We find that the North American indices influence others significantly during the global financial crisis, while during the European sovereign debt crisis, the significant indices are American and European. The US stock indices dominate the world stock market in most periods. The indices of two Asian countries (India and China) influence remarkably in some periods, which occurred due to the unrest state of these markets. The networks based on feature ranking are constructed by assigning a threshold at the mean of the feature ranking matrix. The global reaching centrality of the threshold network is found to increase significantly during the global financial crisis. Finally, we determine Shannon entropy from the probabilities of influence of indices on the target. The sharp drops of entropy are observed during big crises, which are due to the dominance of a few indices in these periods that can be used as a measure of the overall distribution of influences. Through this technique, we identify the indices that are influential in comparison to others, especially during crises, which can be useful to study the contagions of the global stock market.

## 1. Introduction

Globalization of the financial market and its technological advancement have led to a highly competitive stock market internationally. Financial market is a nonlinear dynamical system and needs a holistic view to understand. In the global context, it is more complex as characteristic variables of the market vary in accordance with location. Analyzing the global market helps in estimating worldwide financial risk, measuring dependencies between major global indices, revealing the backend structure of the global market, i.e., finding the connections among indices, finding optimal investment strategies, etc. That's why global financial market analysis using different concepts and methods grabs the interest of researchers. Reconstructing an unknown network structure from the monitored time series has also been a foremost

**Funding:** This work was supported by the ICT Division of Bangladesh under grant number 20FS34427. The funders had no role in study design, data collection and analysis, decision to publish, or preparation of the manuscript.

**Competing interests:** The authors have declared that no competing interests exist.

modern network science problem [1–3]. Many approaches have been used so far to analyze the time series of the global stock market. One such conventional technique that has been used to analyze financial time series in the last two decades is Pearson correlation [4]. Here, from the pair-wise correlation matrix, a threshold network is constructed by assigning a threshold value. When the elements of the correlation matrix exceed the threshold, we keep the element of the matrix; otherwise, we set the element to zero. This way, we obtain the adjacency matrix for the threshold network. In this technique, the correlations between nodes are linear. However, there can exist nonlinear correlations between stocks. Entropy-based mutual information suggests a solution that can handle nonlinear correlation of stocks and hence has been used to analyze financial indices in recent years [5–7]. However, mutual information cannot take multivariate correlation. Besides, both Pearson correlation and mutual information are symmetric, i.e., there is no direction in the network generated by these techniques.

Feature ranking is a technique of Machine Learning (ML). A core learning approach of ML is Supervised Learning (SL), where labeled output or "target" variable is provided along with input "features" in the training data. SL algorithm observes the training data and tries to find relationships and dependencies of the target variable onto the features. The algorithm predicts the output value by taking the features as an argument and compares the predicted value with the corresponding labeled target. From the comparison of the predicted and labeled output, some prediction error is measured according to which the model gets adjusted. Thus, the model learns to predict more accurately. While predicting, the target doesn't depend on all the features to the same degree. Some features have more influence in predicting the target compared to some others. Hence, we can rank features according to their influence on the target. This technique in ML is termed "feature ranking" [8].

In this research, we propose this feature ranking approach of ML as an analyzing tool of the global financial market which can take a multivariate nonlinear view of correlation [9–11]. This approach generates a feature ranking matrix that is asymmetric, i.e., the network generated by this method is directional. Some ML algorithms which integrate this feature ranking method are Decision Tree [12], ReliefF [13], Random Forest [14], and Gradient Boosting [15]. Random Forest [14] is an ensemble learning algorithm that uses a bag of decision trees. Decision tree based algorithms split or partition the data space recursively to make decisions. To do so, it selects the features one by one to perform the split. This selection process is based on an impurity function, e.g., Gini Impurity or Information Gain. To find the best split, Gini Impurity estimates Gini feature importance which is the probability of that feature's contribution in prediction [3]. A feature is assigned with higher importance if the impurity is reduced significantly by selecting it. In every iteration, the Gini feature importance for a data segment is estimated, and the feature with the highest importance is selected to make the split. This way, we find the optimal decision tree and also the rank of features in predicting a target. In Random Forest, decision trees are provided with a random sample of the training data. Each tree in the Random Forest performs the above-mentioned technique on the provided data sample, and the final output is calculated by averaging the predictions of all the trees. XGBoost [15] is another decision tree based ensemble learning algorithm that is very time-efficient. It uses a gradient boosting [16] framework instead of bagging [17] (used in Random Forest). In this research work, we apply both of these algorithms. But the graphs in this article are generated from the feature ranking matrix calculated using Random Forest as they show finer outcomes.

Feature ranking method is a new approach to the financial system. It was first applied to the time series of US stocks in order to understand market structure over time [3]. However, still, there was no study on global financial time series using this technique. To fill these gaps, we apply this feature ranking technique to reconstruct the dynamical network of the global stock

market from the monitored discrete time series. Here, we focus on identifying the influential indices and observing when an index becomes influential in the global market. To do so, we monitor the closing prices of major world indices (nodes) to form our time series dataset. We then consider the dynamical states of a given node on a certain day as the target variable, while the states of all other nodes on the previous day are treated as the features. From this, we measure the influence of each feature on the target, and in this way, the feature ranks for this target are calculated. We use Random Forest [14] and Gradient Boosting [15] algorithms to perform this task. Some features have such a significant influence on the target that we assume a connection from those feature nodes to the target. For low-ranked nodes, we reject their connection with the target node. We set a threshold to filter out insignificant features from the influential features. We use the mean threshold technique in this research to generate the network of influences on the global stock market. To assess the method that we used, we analyzed the topological properties of the network. We also determine Shannon entropy [18] from the probability of influence of features on the target, which gives information on how influences are distributed among indices in a particular period.

## 2. Feature ranking network

### 2.1 Time series formation

Yahoo Finance monitors the daily closing prices of all major world indices [19]. We collected this historical data for 21 important world indices of 21 developed nations (see S1 Appendix) ranging from 2004–2018. This time range includes 3838 days of data. We then divided the time series into 15 disjoint segments using a one-year time window. In each time window, there are around 256 days. During this range, the market encountered three major global crises, such as the Global financial crisis in 2008, the European sovereign debt (ESD) crisis in 2011, and the 2015–2016 stock market sell-off. The daily return $r_i(t)$ of $i^{th}$ global index on day $t$ can be calculated as,

$$r_i(t) = ln[P_i(t)] - ln[P_i(t-1)] \tag{1}$$

where $P_i(t)$ is the closing price of a global index $i$ on day $t$. In this way, we can observe and measure the time series of the dynamics of global indices.

### 2.2 Reconstruction method

We now briefly describe how we use the above-discussed feature ranking approach to analyze the global stock market and reconstruct the market network from the monitored time series. In the network of feature ranking, a node represents the stock index of a country. In each step, we select a node as a target and keep all other nodes (including the target node) as features. We now define a supervised learning algorithm to build our predictive model. The model tries to predict the state of the target node on a certain day using the states of the feature nodes on the previous day. By doing this for each day of a time window, we find the feature importance for that target node in a certain time window. The fact we need to clarify here is that our aim isn't to build a predictive model here, rather, we focus on ranking the features for the target node. Nevertheless, the predictive model is built. In each step, we select a different node as the target node and calculate feature ranks by repeating the same procedure. After doing this for all nodes, we find the feature ranking matrix. In this matrix, highly ranked nodes have a significant influence on the target and are more likely connected to it. To filter out insignificant nodes from the significantly influential nodes, we set a threshold value. This threshold value

determines the links between nodes and thus the network is reconstructed. The fact to be noted is that the feature ranking matrix is asymmetric. So the links here are directional.

Now, to calculate the feature ranking matrix for the current return of an index based on the previous returns of other indices, we formulate the dependence as,

$$r_i(t+1) = f_i(r_1(t), r_2(t), \ldots, r_N(t)), \; i = 1, \ldots, N \tag{2}$$

where $r_i(t+1)$ is the state of node $i$ at time $t+1$, which is calculated from Eq 1. Here, node $i$ is the target node whose state is influenced by the states of all feature nodes $(r_1(t), \ldots, r_N(t))$ at some prior time $t$ and $N$ is the number of nodes. The interaction function $f_i$ is less important here and is unknown, but can be modeled from the time series. We define our training dataset $D_i$ for the target node $i$ for a certain time window with $L-1$ days.

$$D_i = \bigcup_{t=1}^{L-1} (r_1(t), r_2(t), \ldots, r_N(t); \; r_i(t+1)) \tag{3}$$

To calculate the feature ranks for the target node $i$, we need to apply ML algorithm $R$ that supports the feature ranking technique on the dataset $D_i$. $R$ can be any of the above-discussed algorithms like Random Forest [14] and XGBoost [15].

$$(F_{i1}, F_{i2}, \ldots, F_{iN}) = R(D_i) \tag{4}$$

Here, $F_{ij}$ is the estimated influence of node $j$ (stock index $j$) on the target node $i$ (index $i$). Now, by defining the dataset for all $N$ nodes (i.e. $D_1, \ldots, D_N$) and applying algorithm $R$ on each of them, we find feature ranks for all $N$ indices. Putting them together in a matrix form gives us our desired feature ranking matrix $F$ of dimension $N \times N$.

$$F = \begin{pmatrix} F_{11} & F_{12} & \cdots & F_{1N} \\ F_{21} & F_{22} & & \vdots \\ \vdots & & \ddots & \\ F_{N1} & \cdots & & F_{NN} \end{pmatrix} \tag{5}$$

The feature ranking matrix defined above is an asymmetric matrix. Here, entry $F_{ij}$ and $F_{ji}$ are not the same. So, the dependencies of indices are not reversible. This is reasonable since powerful indices may have an exclusive impact on the others. Hence, the network obtained from the feature ranking matrix is intrinsically directional.

To reconstruct the network from the feature ranking matrix $F$, we establish the links between nodes. A link from the node $p$ to $q$ ($p \rightarrow q$) exists if the node $q$ has a significant impact on $p$ i.e. the value $F_{pq}$ is significant. To test whether the impact of a feature is significant or not, we set up a threshold value $\theta$. Thus, we filter out links for low-ranked features. The network structure is sensitive to the threshold. So, when we alter the threshold, the topological properties of the financial network have changed. The topological properties of the network around the mean threshold are finer than other thresholds for understanding the market movement. Hence, we use the mean threshold technique, which can be defined as,

$$\theta = \frac{1}{N * N} \sum_{i=1}^{N} \sum_{j=1}^{N} F_{ij} \tag{6}$$

Finally, we generate the reconstructed adjacency matrix $\hat{A}$ for the network by applying the threshold $\theta$ on the feature ranking matrix $F$.

$$\hat{A}_{ij} = \begin{cases} 0 \ if \ F_{ij} \leq \theta \\ 1 \ if \ F_{ij} > \theta \end{cases} \tag{7}$$

A link $i \rightarrow j$ exists between node $i$ and $j$ if the entry $\hat{A}_{ij}$ is assigned to 1.

## 3. Results and discussions

### 3.1 Feature ranking matrix

The feature ranking matrix is a measure of influences of indices on each other, determined by using machine learning tools. The feature ranking matrix is asymmetrical. Because the influence of a developed market on developing or emergent markets is usually high, equal influence by them may not be possible. Fig 1 shows the feature ranking matrices for three crises of 2008 (global financial crisis), 2011 (ESD crisis), and 2016 (global decline) respectively. To observe the influence of an index on others, let's pick a country on the lower horizontal axis and scan vertically up. The change of color indicates that the influence of an index on others is not equal. For example, during the global financial crisis, the influence of the US on Japan and Australia is remarkably higher than on other countries, shown by white shaded areas in Fig 1A. It implies that the returns of Japan and Australia are significantly affected by the return of the US during the global financial crisis. Similarly, the return of the index of the US is influenced remarkably by Canada. We observe that during the global financial crisis, Asian and European indices are affected significantly by the indices of American countries. It implies that American influences dominate the world in this period. During the ESD crisis, Asian indices are influenced by the indices of American and European shown in Fig 1B. In most periods, Asian markets are found to be more sensitive to other continental markets. Perhaps they are not as developed as American and European stock markets. We need further analysis in this regard. Each index can influence its own. Here, we see that the return of the index in Japan is remarkably influenced by its own. Again, during the period of the global decline in the value of stock prices that occurred between June 2015 and June 2016, the returns of most indices are controlled by the US, shown in Fig 1C. The second influential country is Canada. The index of Israel is significantly affected by the US in this period. From the time series, we observe that the returns of any index can change at random or follow some pattern. When they change randomly, self-influence remains low compared to the influence of other indices. But self-influence is high when returns follow some pattern. It is observable from the main diagonal of the feature ranking matrices.

### 3.2 Influence of individual indices

The dynamic influence of an index on others is shown in Fig 2. The total influence $F_{tot}$ of an index $j$ is given as,

$$F_{tot}(j) = \sum_{i=1}^{N} F_{ij}. \tag{8}$$

We only analyze those indices that have a remarkable influence on others. Fig 2A shows that the influence of the US is higher than other countries over time. The significant influence of this index is found during different financial crises and its influence becomes lower just before crises. For example, the influence of the US index declined in 2006 just before the global

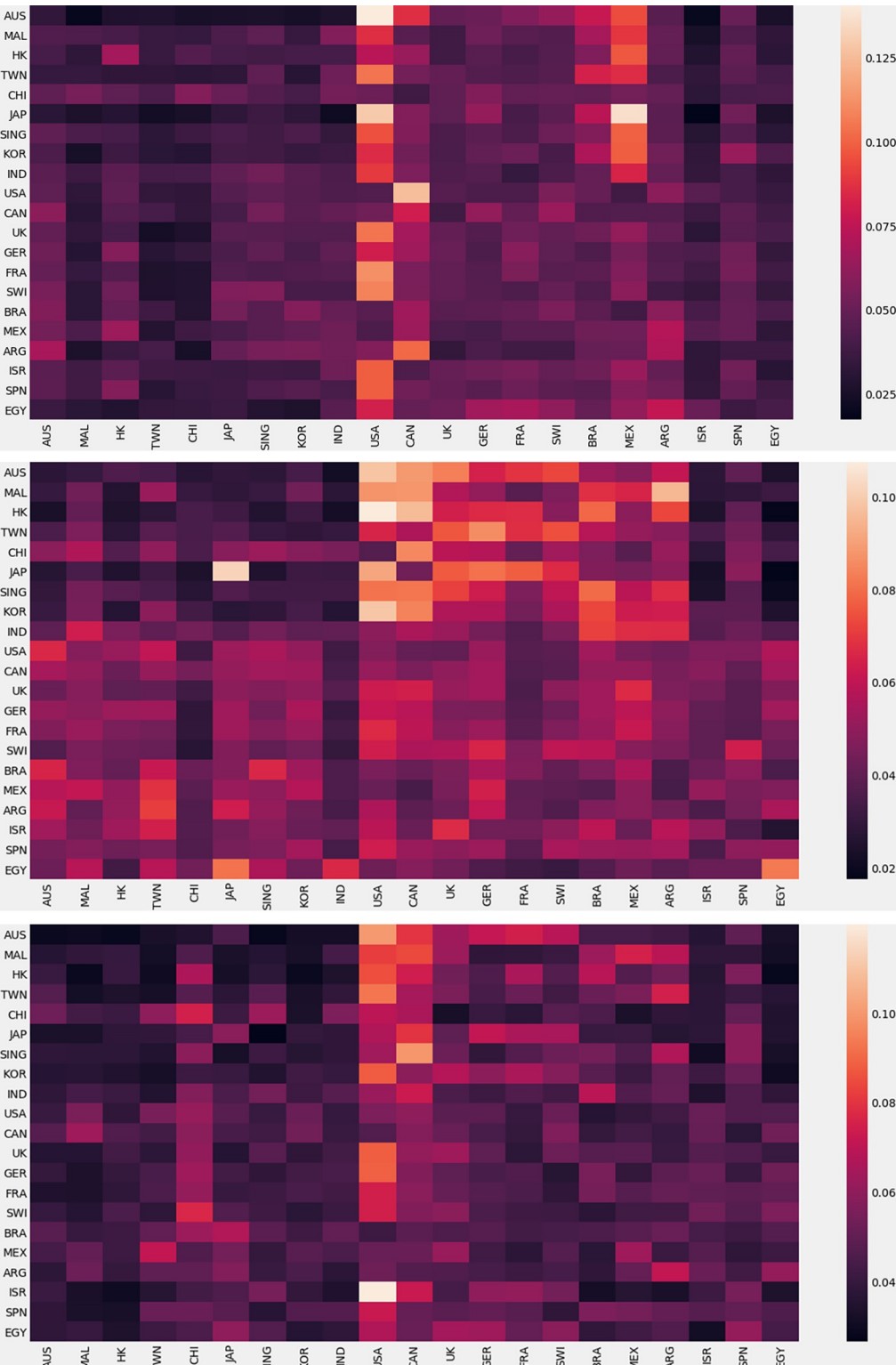

**Fig 1. The feature ranking matrices for 21 global indices.** (a) during the global crisis (2008), (b) during the ESD crisis (2011), (c) during the period of the global decline in the value of stock prices (2016). Lighter color indicates higher influence and darker color specifies the opposite.

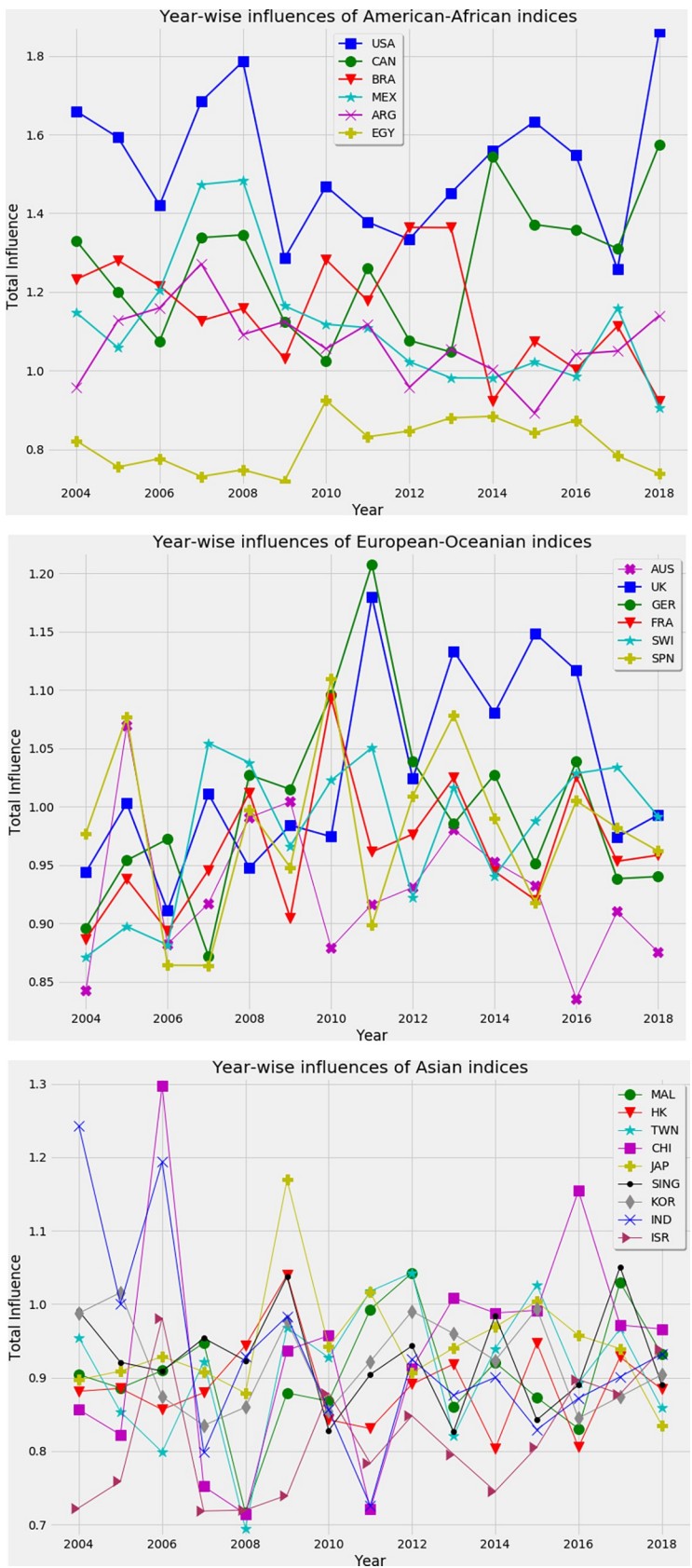

**Fig 2. The dynamics of total influences of an index to others.** (a) American-African indices, (b) European-Oceanian indices, and (c) Asian indices. The dominance of the US index is observed in most of the periods. The index of China and India influence other indices in some periods due to the unrest state of these markets.

financial crisis, in 2009 just before the ESD crisis, and in 2017 just before stock prices fall in 2018. The index of Canada also has a significant influence during financial crises. But, the remarkable influence of this index is found after 2013. From Fig 2A, it is also noticeable that in 2018, the influence of the US and Canada rise sharply while other indices possess lower influence. This implies that most part of the influence is confined to these two indices in this period. The influence of the Brazilian index is seen to decline from 2004 to 2008 and after that, its influence increases sharply during the ESD crisis in 2010 and the Cypriot crisis in 2012–2013. After that, the dominance of this index decreases. The significant impact of the Mexican index is observed around the global financial crisis.

Next, we consider European countries, and we found that most of the indices are dominant in some periods as shown in Fig 2B. For example, the influence of the UK rose sharply from 2011 with high fluctuation and decreased in 2017–2018. The remarkable influence of Germany is found during the ESD crisis in 2011. The moderate influence is seen for the indices of France and Spain in some periods of our analyzing time. When we look for the influence of Asian indices in Fig 2C, we observe that the dominant indices are for the countries of India, China, and Japan in some periods. In 2004, the index of India falls 15.52%, and it is the largest fall in history in terms of percentage, which influences other indices in the world. Similarly, the BSE Sensex of India fall by 826 points in May 2006 and consequently, a significant influence of it is found this year. On the other hand, the remarkable influence of China's index is found in 2006 and 2016, which is due to the bubble of the market in these periods. The significant impact of Japan's index is found in 2009 due to the effect of the global financial crisis. In summary, we find that when a stock market falls into a crisis, its impact is found in other markets as well. This kind of influence can be identified by the technique of feature raking of machine learning.

### 3.3 Network construction

The feature ranking networks are constructed by assigning a threshold at the mean of elements of the feature ranking matrix. Two indices are connected by a link if the feature importance is higher than the threshold $\theta$. The mean threshold of the feature ranking matrix for 21 global indices is $\theta = 0.047$. When feature importance $F_{ij} > \theta$, where $i, j = 1, 2...,N$, a link will be added in the financial network connecting node $i$ to node $j$. The global financial network for the year 2008, when the financial crisis ignited all over the world is shown in Fig 3. The direction indicates that the source index is influenced by the destination index and some arrows are bi-directional, which implies both indices are influencing each other. For example, the index of the US influences 18 indices all over the world during the global financial crisis. The indices of the US and Switzerland influence each other because the links are bi-directional as shown in Fig 3. On the other hand, we observed that some Asian countries like Taiwan do not have any in-degree, which means those countries have no substantial influence on any other country in the network. The dominance of American indices is higher than other indices during the global financial crisis. The index of China has no influence on others in this period. The indices of Singapore, Japan, and South Korea have little influence. The indices of Europe don't contribute that much. These indices influence others moderately in this period. The index of Israel influences only the index of Egypt. This kind of network structure is useful to observe the dominance and interaction of indices and to estimate which country influences whom in the global market.

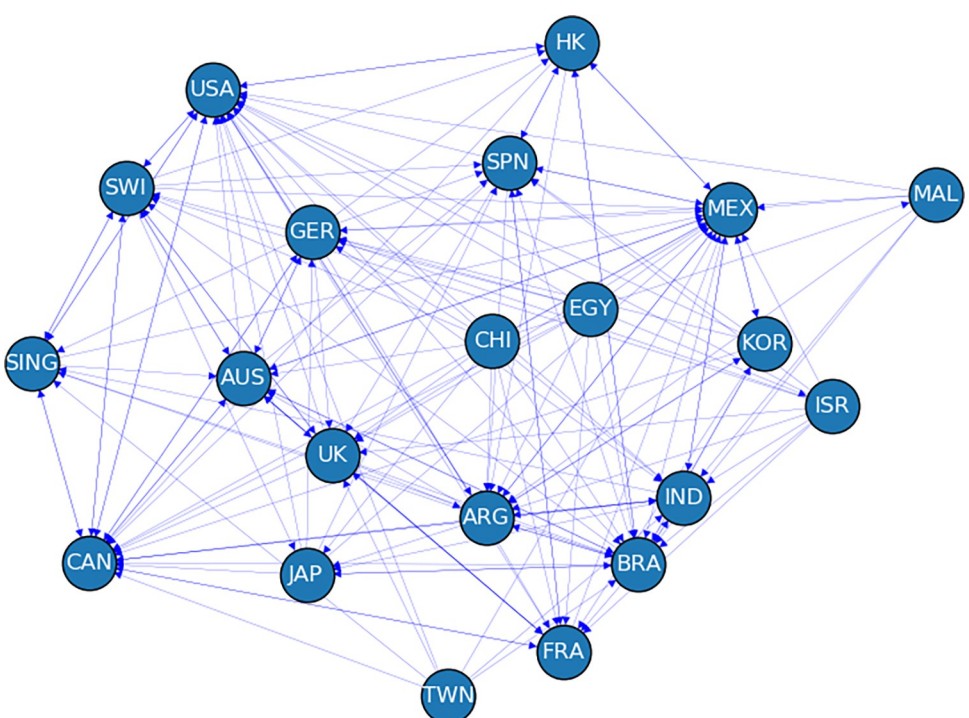

**Fig 3. The global financial network of influence is constructed by the mean threshold during the global financial crisis (2008).** Unequal influences are observed among the indices. American indices influence more than other indices. Full forms of node labels are given in the S1 Appendix.

## 3.4 Significant indices in the network

The dynamics of the number of influenced indices are shown in Fig 4. This figure is useful to show how many indices are influenced by one another in a period. The number of influenced indices, $I_N$ by an index $j$ is calculated as,

$$I_N(j) = \sum_{i=1}^{N} \hat{A}_{ij} \tag{9}$$

where $\hat{A}_{ij}$ is the reconstructed adjacency matrix of the feature ranking network and $N$ is the number of nodes. Here, $I_N(j)$ is the number of in-degree for the node $j$ because $\hat{A}_{ij}$ indicates that $i(target) \rightarrow j$, as $F_{ij} > \theta$. Let's analyze the dynamics of the number of influenced indices. First, we consider the American indices in Fig 4A. The number of indices that are influenced by the US is higher than others but fluctuates over time. The dominance of the US index declines remarkably in 2012. After that, the influence again increases over time, and during the global decline of stock prices in 2015, it influences all 21 indices. The next dominant stock index is the Canadian index. The influence of this index fluctuates remarkably over time. Similarly, the influence of other American indices varied over time. After 2013, the dominance of the US and Canada increase while the influence of Brazil, Mexico, and Argentina decrease.

Next, we consider European indices as shown in Fig 4B. The influences of European indices are comparatively less than American indices and fluctuate over time. The significant influences of these indices are found during the ESD crisis in 2010–2011. The dominating indices are the index of Germany, France, Spain, and the UK. The influence of Germany increases from 2008 and reaches at peak in 2011. After that, the influence of this index declines remarkably. The dominance of the UK index is found from 2011 to 2016. After the ESD crisis, the

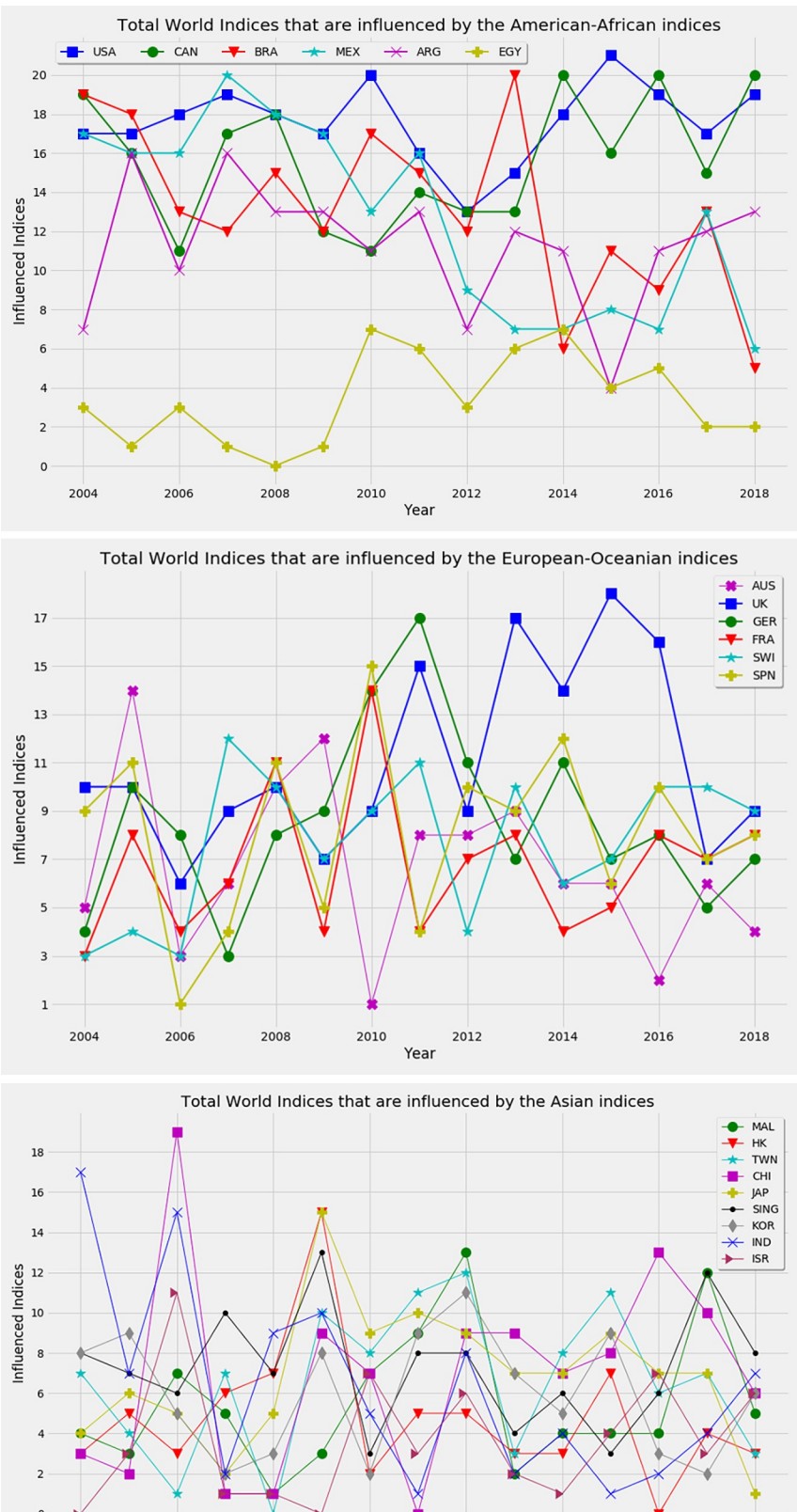

**Fig 4. The number of indices that are influenced by other.** (a) American-African indices, (b) European-Oceanian indices, and (c) Asian indices. The dominance of the US index is found in most of the periods.

influences of most European indices decrease. The influence of Asian indices is less than American and European indices. The most influential indices are the index of India, China, Singapore, Japan, Malaysia, and Hong Kong in some periods. Although the markets of China and India are emerging, they influence others significantly in some periods as shown in Fig 4C, due to the unstable state of these markets. After the global financial crisis, the dominance of the markets of Singapore, Japan, and Hong Kong is observed in 2009. The supremacy of the Malaysian index is observed in 2012 and 2017. In 2018, a noteworthy number of indices are influenced by the US and Canadian indices compared to the others, indicating the exclusive dominance of these two indices in this period. The rest of the indices influence others moderately during our observation. In summary, American indices influence more than other indices. The dominance of the US index is found in most of the periods. When a market falls into a crisis, it influences other markets. For example, American indices influence much more during the global financial crisis in 2008 while the dominance of European indices is seen in 2010–2011. Again, the significant influences of two emergent markets like China and India are found in 2004 and 2006, respectively, due to the turbulence of these markets.

### 3.5 Topological properties

**3.5.1 Network density.** The network density is the ratio of the number of existing links to the maximum number of possible links, which can be determined for the directed graph as [20, 21],

$$\rho = \frac{M}{[N(N-1)]} \tag{10}$$

where $N$ is the total number of nodes and $M$ is the number of connecting links. Here, bi-direction is possible; hence the maximum number of possible links is $N(N-1)$. The network density of the threshold network of the global stock market at the mean threshold $\theta = 0.047$ is shown in Fig 5A. The variation of network density can be used to identify the financial crisis. The network density sharply increases around the global financial crises in 2007–2008, ESD crises in 2010–2011, and during the global decline of stock prices in 2015–2016. There is an additional peak in 2005. The indices influence each other more in these periods. Now, let's consider why network density is lower during global financial crises than during other crises. This is mainly for the dominance of American indices in this period, shown in Figs 3 and 4. The unequal influences among the indices generate lower network density. The higher network density is due to the greater influence of more indices in the network, which is observed around the ESD crisis. After that, the network density decreases and again increases due to the global decline of stock prices in 2015–2016. A sharp drop in network density is found in 2018 due to the exclusive dominance of the US and Canadian indices shown in Figs 2A and 4A. The dynamic change of network density is the reflection of influences among the indices.

**3.5.2 Average shortest path.** The average shortest path length is an average of the shortest path lengths between every pair of nodes in the graph. The characteristic path length or the average shortest path length in a cluster can be expressed as [3, 22],

$$\bar{l} = \frac{1}{N(N-1)} \sum i, j \, d_{ij} \tag{11}$$

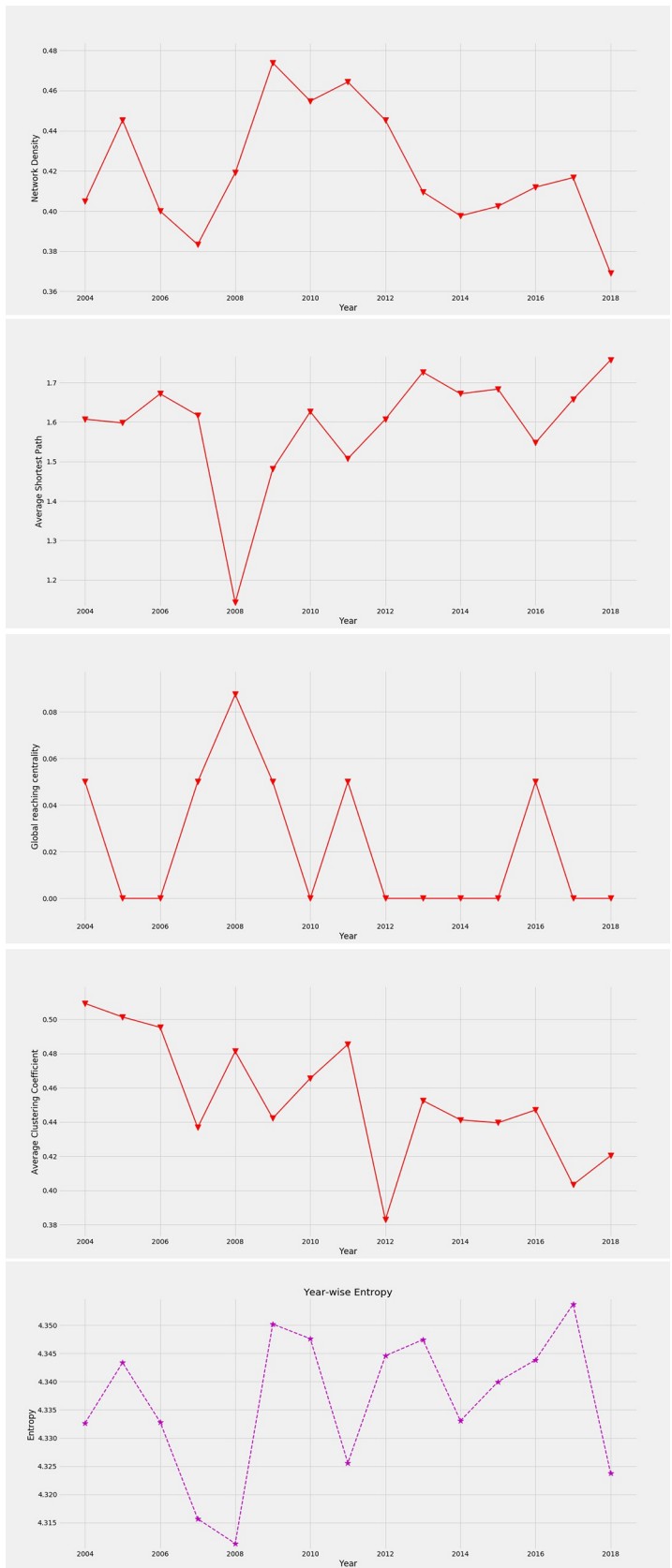

**Fig 5. Topological properties of the threshold network at the mean threshold.** (a) network density, (b) average shortest path length, (c) global reaching centrality, (d) average clustering coefficient, and (e) entropy of the feature ranking matrix.

where $d_{ij}$ is the shortest path length between nodes $i$ and $j$. The average shortest path length of the threshold network of the global market at threshold $\theta = 0.047$ is shown in Fig 5B. The average shortest path of the global financial network of influence fluctuates over time. This variation is evident and notable due to financial crises. A sharp drop in it is observed during the global financial crisis in 2008, which indicates that more indices are influenced by American indices in this period. However, the change in the shortest path during the ESD crisis in 2011 and the global decline of stock prices in 2016 is mild. It indicates that the global financial crisis was different from other crises and more significant. The largest distance is observed in 2018 when the amount of link distribution of other indices is noticeably less than the two American giants (US and Canada) shown in Fig 4A.

**3.5.3 Global reaching centrality.** The global reaching centrality (GRC) is a global network quantity that shows the flow hierarchy of a complex network, where nodes contribute differently to the dynamics of the network. It is defined for the directed graph as [3, 23],

$$GRC = \frac{\sum_{i \in V}[C^{max} - C(i)]}{N - 1} \tag{12}$$

where $C(i)$ is the local reaching centrality ($LRC$) of node $i$ depicting the proportion of all nodes in the network that can be reached from node $i$ via outgoing links and $C^{max}$ is the maximum value of $LRC$, $V$ is the set of vertices. The global reaching centrality of the financial threshold network at the mean threshold $\theta = 0.047$ is shown in Fig 5C. The figure shows that the curve only lifted during different financial crises from 2004 to 2018. The curve reaches its peak in 2008, indicating the maximal heterogeneous distribution of the $LRC$, which implies a maximal hierarchical state of the market during the global financial crisis in 2008. But during other financial crises such as the historical collapse of the BSE Sensex in 2004, the subprime mortgage crisis in 2007 and 2009, the ESD crisis in 2011, and the global decline in the value of stock prices in 2016, the GRC keeps the same value, which means that the network possesses the same level of hierarchy in these years. The increase of the GRC depicts that there is a central portion of the network which exclusively influences the maximum part of the network directly or via multiple hops. In the remaining years, the GRC does not lift at all, which means that the numbers of nodes that are reachable from all the nodes individually in the network, directly or through multi-hop, are identical. The $LRC$ of all the nodes are identical, and hence there is no notion of the hierarchy during these stable states.

**3.5.4 Average clustering coefficient.** The average clustering coefficient (ACC) is a measure of the compactness and how well a network is aggregated. The local clustering coefficient of a vertex $i$ is defined as [21, 22, 24],

$$C_i = \frac{m_i}{n_i(n_i - 1)} \tag{13}$$

where $n_i$ denotes the number of neighbors of vertex $i$, and $m_i$ is the number of edges existing between the neighbors of vertex $i$. $C_i$ is equivalent to 0 if $n_i \leq 2$. Then, we can calculate the average clustering coefficient $C$ for the entire network by simply taking the average of the local

clustering coefficients of all nodes as follows,

$$C = \frac{1}{N}\sum\nolimits_{i=1}^{N} C_i \tag{14}$$

The average clustering coefficient of the threshold network of the global stock market is shown in Fig 5D at the mean threshold. The ACC starts with the highest value in 2004 in Fig 5D, but the corresponding network density is low during this period. So a group of compact local clusters is formed around some influential nodes for which the ACC starts with the highest value during the BSE Sensex crash in 2004. But compactness of some local clusters reduces and cross-cluster communication rises over time. Some local clusters break at the beginning of the subprime mortgage crisis in 2007. Therefore, both the density and the average clustering of the network get reduced. Nodes of these broken clusters get connected to the major clusters during the global financial crisis in 2008 and the ESD crisis in 2011 and hence showed a rising trend during these periods. But as some nodes lose their local compactness, the trend can't exceed the average clustering coefficient of 2004–2006. The major clusters during the global financial crisis are American, and during the ESD crises, they are American and European, as they are the most influential during these periods, which we find in Figs 2 and 4. We also observe an increasing trend during the decline in the value of stock prices in 2016. A sharp drop is noticed in 2012, just after the ESD crisis.

### 3.6 Entropy

Entropy is a measure of uncertainty of a random variable that takes probability values. As the feature importance of the feature ranking matrix represents the probability of contribution of a stock index in predicting the target index, hence we can calculate entropy from the feature ranking matrix. The probability of contribution is distributed among 21 stock indices of 21 different countries around the world. So, the total probability of contributions in predicting a target is one. The entropy is calculated as [25],

$$S = -\frac{1}{N}\sum\nolimits_{i,j=1}^{N} F_{ij} log_2(F_{ij}) \tag{15}$$

where $F_{ij}$ is the element of the feature ranking matrix $F$ at row $i$ and column $j$. The entropy will be at its highest value when the indices influence each other equally. The variation of entropy with time is shown in Fig 5E. It indicates the unequal influences among the indices in unstable periods. We found that the entropy decreases before and during two big crises in 2008 and 2011. It emphasizes that a few indices influence most of the indices in these periods. The main reason behind it is the instability of the market. When a market falls into a crisis, it spreads to other markets, and hence, the index leads the world market in that period. However, some developed markets lead the world market during the normal period as well. For example, a sharp drop in entropy is noticeable in 2018. The reason behind this is the exclusive influence of two American giants (US and Canada) in this period which leads to an unequal influence of indices that is observable from Fig 2A. In a mild crisis like the global decline of stock indices in 2015–2016, the entropy increases before and during crises. It indicates that the influences among the indices are becoming equal. Hence, the entropy can be an indicator to observe the overall distribution of influences in the global market.

## 4. Conclusion

The feature ranking approach provides information about the influence of an index on the target. We found that American indices influence others significantly during the global financial

crisis. While during the ESD crisis, American and European indices are most influential. We also found that two Asian indices, China and India, influence others during the turbulence of these markets. The index of the US is dominating in most of the periods, and it influences all indices in 2015 when the prices of the global stock indices are declined. We construct networks at the mean threshold. The network structure can identify the interaction among indices. China has no influence on others during the global financial crisis, as observed in the network. The topological properties of the financial network of influence are studied over time. The network density and average shortest path drop sharply before and during the global financial crisis. The global reaching centrality is higher during the global financial crisis than in other crises. It indicates that the network of influence is maximally hierarchical in this period. Finally, the Shannon entropy of the probability of influences among indices is shown. The sharp drop in entropy during two big crises indicates unequal influences among the indices.

Financial crisis is the most important phenomenon in the financial market. The feature ranking approach can be a proper technique to study the crisis and identify the influential indices, especially during crises. Applying this method to other financial time series may be useful and interesting in understanding the contagion in the financial system.

## Supporting information

**S1 Appendix. Stock markets observed in this research.**
(DOCX)

## Acknowledgments

The author thanks Prof. Jae Woo Lee for his valuable comments.

## Author Contributions

**Conceptualization:** Ashadun Nobi.

**Data curation:** Mahmudul Islam Rakib.

**Formal analysis:** Mahmudul Islam Rakib, Ashadun Nobi.

**Funding acquisition:** Mahmudul Islam Rakib.

**Investigation:** Mahmudul Islam Rakib, Ashadun Nobi.

**Methodology:** Mahmudul Islam Rakib.

**Resources:** Mahmudul Islam Rakib.

**Software:** Mahmudul Islam Rakib.

**Supervision:** Ashadun Nobi.

**Validation:** Md. Javed Hossain, Ashadun Nobi.

**Visualization:** Mahmudul Islam Rakib.

**Writing – original draft:** Mahmudul Islam Rakib.

**Writing – review & editing:** Mahmudul Islam Rakib, Md. Javed Hossain, Ashadun Nobi.

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
