## [Decision Letter · Decision Letter 0]

20 Apr 2022

PONE-D-22-06124Feature ranking and network analysis of global financial indicesPLOS ONE

Dear Dr. Nobi,

Thank you for submitting your manuscript to PLOS ONE. After careful consideration, we feel that it has merit but does not fully meet PLOS ONE’s publication criteria as it currently stands. Therefore, we invite you to submit a revised version of the manuscript that addresses the points raised during the review process.

Please, address all the issues and comments by the referee.

We look forward to receiving your revised manuscript.

Kind regards,

Petre Caraiani

Academic Editor

PLOS ONE

Journal Requirements:

Whilst you may use any professional scientific editing service of your choice, PLOS has partnered with both American Journal Experts (AJE) and Editage to provide discounted services to PLOS authors. Both organizations have experience helping authors meet PLOS guidelines and can provide language editing, translation, manuscript formatting, and figure formatting to ensure your manuscript meets our submission guidelines. To take advantage of our partnership with AJE, visit the AJE website (http://aje.com/go/plos) for a 15% discount off AJE services. To take advantage of our partnership with Editage, visit the Editage website (www.editage.com) and enter referral code PLOSEDIT for a 15% discount off Editage services.  If the PLOS editorial team finds any language issues in text that either AJE or Editage has edited, the service provider will re-edit the text for free.

A clean copy of the edited manuscript (uploaded as the new *manuscript* file).

[This work was supported by the ICT Division of Bangladesh under grant number 20FS34427. The author thanks Prof. Jae Woo Lee for his valuable comments.]

 [NO. The funders had no role in study design, data collection and analysis, decision to

publish, or preparation of the manuscript.]

6. We note that you have indicated that data from this study are available upon request. PLOS only allows data to be available upon request if there are legal or ethical restrictions on sharing data publicly. For more information on unacceptable data access restrictions, please see http://journals.plos.org/plosone/s/data-availability#loc-unacceptable-data-access-restrictions. 

Reviewers' comments:

Reviewer's Responses to Questions

**Comments to the Author**

1. Is the manuscript technically sound, and do the data support the conclusions?

Reviewer #1: Yes

2. Has the statistical analysis been performed appropriately and rigorously? 

Reviewer #1: Yes

3. Have the authors made all data underlying the findings in their manuscript fully available?

Reviewer #1: Yes

4. Is the manuscript presented in an intelligible fashion and written in standard English?

Reviewer #1: Yes

5. Review Comments to the Author

Reviewer #1: Reviewer’s comments

This article is applying the feature ranking method to investigate properties of world stock indices.

They identified influential communities during the global financial crisis. The US stock indices dominate the world stock market in most periods. However, they reported that China and India become remarkably influential in some periods. This article is interesting and extends our understanding for the stock market based on the complex systems. I recommend publishing. I include minor comments.

Minor Comments

1. In Eq. 6 you used the average value of elements of feature ranking matrix. Do you have a reason to choose the threshold as the average? Is it a rule of sum? You can give some comments when you change the threshold.

2. When you consider 2008, 2011, 2016 crisis, you use different length of the time series. For example, when you consider 2008 crisis, you use the previous time series before year 2008. Could you give the length of the time series for the training set? Is it the one-year time window to get the feature ranking matrix?

3. In your definition, the self-influence F_ii is determined by the current return influenced by the previous return itself. Could you explain that the self-influence is small compared to other indices?

4. In Fig. 1b, you consider ESD crisis. Then, many Asian countries influenced by American and European countries. Could you explain why Asian countries are so sensitive to other markets.

5. When you define the total influence in Eq. 8, it seems that the F_ij is unnormalized because F_tot (j) varies in a wide range. The absolute value of each F_ij has some meanings?

6. Is there any reason that many Asian countries in Fig. 3 don’t have in-degree?

7. In line 264 below Eq (9), “the adjacency matrix of the feature ranking network….” have to change as “the reconstructed adjacency matrix of the feature ranking network ….”.

8. In Eq. 9, I_N (j) is the number of out-degree for the node j because hat{A}_ij indicate that j � i(target) as hat{A}_ij >= theta.

9. Do you neglect the direction when you define the network density, average shortest path?

10. In Eq.12, you defined the local reaching centrality. Then it means that all possible nodes you follow out-going links starting from node i. For example, the node i�j�k�l, then the node l is reachable from i to l. Do you mean this type of reaching? Could you explain clearly?

11. Check typos.

6. PLOS authors have the option to publish the peer review history of their article (what does this mean?). If published, this will include your full peer review and any attached files.

Reviewer #1: No

---

## [Author Response · Author response to Decision Letter 0]

27 Apr 2022

Reviewer 1: We have incorporated your suggestions into our revision. They were very helpful. Thank you.

---

## [Editor Report · Decision Letter 1]

23 May 2022

Feature ranking and network analysis of global financial indices

PONE-D-22-06124R1

Dear Dr. Nobi,

We’re pleased to inform you that your manuscript has been judged scientifically suitable for publication and will be formally accepted for publication once it meets all outstanding technical requirements.

Kind regards,

Petre Caraiani

Academic Editor

PLOS ONE
---

## [Editor Report · Acceptance letter]

25 May 2022

PONE-D-22-06124R1 

Feature ranking and network analysis of global financial indices 

Dear Dr. Nobi:

I'm pleased to inform you that your manuscript has been deemed suitable for publication in PLOS ONE. Congratulations! Your manuscript is now with our production department. 

Kind regards, 

on behalf of

Dr. Petre Caraiani 

Academic Editor

PLOS ONE